# Do chimpanzees (*Pan troglodytes*) mentally represent collaboration?: Action-learning and communication in a partnered task

**Elizabeth Warren**[1]☉*, **Emma Suvi McEwen**[2]☉*, **Josep Call**[2]

1 Johns Hopkins University, Baltimore, Maryland, United States of America, 2 University of St Andrews, St Andrews, Scotland

☉ EW and ESM contributed equally to the manuscript
* ewarren8@jh.edu (EW); esm7@st-andrews.ac.uk (ESM)

## Abstract

Non-human primates engage in complex collective behaviours, but existing research does not paint a clear picture of what individuals cognitively represent when they act together. This study investigates chimpanzees' capacity for co-representation. If individuals represent others' actions as they relate to their own during a collaborative task, they should more easily learn to reproduce that action when their roles are switched. In a between-subjects design, we trained ten chimpanzees (*Pan troglodytes*) on a sequential task, in which the first action is performed by either a human partner or a non-social object, and the second action is performed by the subject. We then imposed a breakdown in the action sequence, in which subjects could perform both actions themselves, but received no help from the experimenter or object. We measured subjects' success in reproducing the first action in the sequence, as well as their attempts to recruit the experimenter's help using requesting gestures. We found no overall difference in subjects' ability to perform the first action in the sequence, but we observed significant qualitative differences in their solutions: individuals in the partnered condition replicated the experimenter's action, while those in the non-social condition achieved the same end using alternative methods. This difference in solution style could indicate that only those chimpanzees in the partnered condition mentally represented the experimenter's action during the collaborative task. We caution, however, that given the small number of subjects who solved the task, this result could also be driven by individual differences. We also found that subjects consistently produced communicative gestures toward the experimenter, but were more likely to do so after exhausting all actions they could take alone. We suggest that these patterns of behaviour highlight a number of key empirical considerations for the study of coordination in non-human primates.

**Data availability statement:** All data and video files associated with this manuscript can be found in the following OSF repository: https://osf.io/sdj7w/?view_only=a8e6155134794723a5ad1e1d2481785b

**Funding:** This work was supported by the European Union's Seventh Q20 Framework Programme (FP7/2007–2013)/European Research Council Grant 609819 (SOMICS) to author JC. The funders had no role in study design, data collection and analysis, decision to publish, or preparation of the manuscript.

**Competing interests:** The authors have declared that no competing interests exist.

## Introduction

Collaboration between individuals is fundamental to the survival and success of many group-living species. Successful coordination toward a common or complementary goal can be observed in a wide range of animal behaviours, for instance: collective foraging and hunting, group navigation, territory defence, and also the elaborate systems of collaboration undertaken in human societies. Non-human great apes (hereafter, "apes") are known to engage in a number of complex collective behaviours, such as co-travel [1–3], coalition defence [4,5], cooperative food-sharing [6], and collective hunting [7,8]. Non-human primate (hereafter, "primate") collective behaviours suggest some cognitive ability to interpret and react toward others' behaviour in a goal-oriented way, but it is a challenge to assess exactly what is known to each individual and what each agent represents about the interaction [9,10]. The study of collaboration in primates opens a window into the evolution of cognitive mechanisms such as the ability to represent the actions and goals of others, and the ability to communicate to recruit and instruct a partner. These same cognitive mechanisms are the building blocks of an evolutionary progression toward more complex collective activities seen in humans, such as cultural practices, teaching, and advanced systems of leadership and governance [11,12]. Here, we investigate chimpanzees' action learning and communication behaviours during an interspecific collaborative task, to explore the species capacity for co-representation and the cognitive mechanisms at play in their collective behaviour.

Collective behaviour can involve group activity at varying levels of cognitive complexity, and may or may not include deliberate coordination between individuals. Here we define collaboration as an interaction in which two or more agents act on the same task, producing identical or complementary actions toward a single outcome. Collaboration may or may not include coordination, in which one or more of the agents cognitively represents how each others' actions relate to the goal ([9] "actively coordinated collaboration"). This representation need not be symmetrical, one partner may use the other as a social tool, and the latter need not engage in coordination on a cognitive level. Often in studies of social coordination with primates, it can be challenging to determine what subjects are representing about the task. Individuals could each be focused only on their own role and incidentally work together [13]. Many would argue that this scenario does not reach the criteria for coordination [14,15]: "planned coordination"). More cognitively complex collaboration might include, for example, shared intentionality, in which both collaborating partners have a representation of their joint motivation to act together [9,16].

One criterion that has been used to detect coordination in human adults is the presence of "co-representation", referring to when participants not only represent their own task but also aspects of their task partner's intentions, goals, or actions [17,18]. Recent investigations across several monkey species have suggested that some primates engage in co-representation [19,20]. These studies employed a 'Joint Simon' paradigm, wherein subjects must respond to one cue, with a conspecific partner present who must respond to a different cue [18,21]. In this paradigm, the authors identify co-representation when an individual's performance declines

due to interference from their partner's cue. Each individual could solve the task by attending only to their own cue and ignoring the other, and subjects do successfully inhibit their responses to irrelevant cues in solo tasks. Failure to inhibit responses to the other's cue in "joint" conditions, therefore, may suggest representation of the partner's task, in addition to one's own. While these studies report evidence of co-representation in primates, the authors themselves point out that the paradigm has certain pitfalls. Namely, successful co-representation negatively impacted task performance; subjects fared better when they could inhibit their representation of the others' role. The authors suggest that a clearer diagnostic for co-representation would be a task that does not demand its inhibition [22]. The Joint Simon paradigm, particularly in its adaptations for primates, also has limitations from other factors that may have confounded the monkeys' response inhibition. The monkeys may not have understood that they should not respond to their partner's cue, and their lack of inhibition could reflect a failure to understand that the task is actually split between two agents [23]. Furthermore, as both subjects receive the reward when either responds correctly, subjects may respond to their partner's cue not because they are representing their partner's task, but because they have their own conditioned association between that cue and a reward. Therefore, it is important to find alternative paradigms to the Joint Simon task for probing co-representation in primates.

An alternative to the Joint Simon paradigm is to look at the result of co-representation on subsequent action learning, rather than *in situ* interference with performance. Successful co-representation may provide an opportunity to learn actions from a social partner, scaffolded by a joint goal. Chimpanzees have demonstrated some social learning abilities; they can learn actions through observation (for example, nut-cracking: [24], manipulating experimental puzzle boxes: [25,26]). While there is some disagreement over the degree to which behaviour can be socially learned in non-human animals [27,28], social learning has been reported across several species [29]. The social transmission of behaviour in chimpanzees could be evidence that, in some contexts, chimpanzees possess a system for representing (and hence learning from and sometimes reproducing) other's actions. Whether or not this is a faculty also co-opted for coordination tasks remains to be answered. Social learning and coordination are connected phenomena and research on the two topics share focuses such as mapping another's actions onto one's own motor system (to learn from, or to make predictions about actions), and working jointly may aid information transmission [30,31]. Previously, in a joint task with two complementary roles, [32] found that chimpanzees learned an action 50% faster after reversing roles than those who learned the role initially, although this effect was not significant. [10], however, points out that the actions required in this task were relatively complex, which may have made vicarious action learning particularly challenging with this apparatus. Given that [32] did find a pattern in the expected direction, it is plausible that an alternative paradigm which includes more training in the relevant task demands could more accurately assess action learning differences. Additional measures, such as rate of communication behaviours, may indicate subjects' expectations of their partner and offer further insight into apes' representations of the collaborative nature of a task.

The evolutionary relationship between collaboration and communication has been explored in depth in both humans and primates. It has been suggested that, in humans, the evolutionary pressure to collaborate may have driven the phylogenetic development of increasingly complex social cognition and further facilitated the evolution of language [11,33–35]. According to that view, communication and language expanded and specialised as necessary means by which to coordinate actions and attention toward common goals (although see [36] for an alternative view of the evolution of shared intentionality). The intertwined connection between communication and collaboration is of interest in the study of primate cognition, not only for the purposes of tracing the evolutionary roots of human communicative abilities, but also because primates engage in stable collective behaviours in captivity and in the wild (e.g., collective hunting, group travel). Successful coordination and collaboration in primates allows for the possibility that mechanisms of flexible, goal-directed communication are at work, but the extent of primates' ability to act on any shared representation of goals through the route of communication is still not well understood. The link between collaboration and communication indicates that communication may be an appropriate measure of shared representation during coordination tasks. Communication to reestablish coordination when it breaks down would suggest that the communicating agent has formed some representation of the relevance of their partner to the task.

Several studies have investigated whether primates, and especially great apes, use communication to establish, sustain, or re-initiate collaboration tasks (see [37] for review), but with inconsistent or inconclusive findings. Several experiments have found evidence for non-communicative social tool use or manipulation, such as passively waiting at a collaborative apparatus for a partner to engage (e.g., [38]), or physically manipulating a conspecific to approach or act on an apparatus (e.g., [39,40]). Other investigations, however, do find that apes will use "attention-getting" vocalisations and gestures while waiting in proximity to a collaborative apparatus [41,42]. In stag-hunt paradigms, chimpanzee pairs tend to opt out of collaboration when communication is required to initiate it, but do successfully coordinate when they can use a more passive leader-follower strategy [43,44]. In the existing literature, great ape communication to establish coordination tends to take the form of physical manipulation and attention-getters, when it occurs at all.

Regarding communication to re-establish collaboration, the evidence is likewise mixed. Warneken and colleagues [45] found that infant chimpanzees did not communicate to re-engage a human partner in a social game, instead inventing and persisting in a solo version of the task. Conversely, subsequent research has shown that chimpanzees are sensitive to coordination breakdowns and do produce some gestures to re-engage task partners (for instance, [46,47,48]). In their review, [37] outline certain factors that influence the production and reception of communication in the context of collaboration. Critically, they find that an asymmetry of knowledge between the collaborating partners, such that to achieve the goal, one must necessarily recruit the attention of or transmit information to the other, is the most likely scenario to induce communication. [37] raise the point that an investigation juxtaposing the attentional or knowledge state of the recipient – whether they are known to have the necessary information to perform their role or not – could probe the use of communication to establish coordination. If subjects communicate more often to engage a temporarily disengaged partner with whom they have a history of successful coordination, it might suggest that they have represented the act of coordination and are intentionally trying to recruit their partner back to the task. From the perspective of co-representation, it may be that the communication produced in asymmetrical- attention/knowledge task structures indicates the producer's representation of their partner's necessary role in the task. This existing literature suggests that the metric of communication is a tool rich with potential to parse primates' representations of partners' roles. The producer's rate of communication under different task parameters may give clues as to their representation of their partner's relevance, knowledge state, and motivation to collaborate on the task at hand.

In the current study, chimpanzees worked to obtain a food reward from an apparatus in a sequential, two-action task. Over the course of a learning phase, subjects experienced the task either as collaborative, in which the first action in the sequence was performed by a human (partnered condition), or as mechanical, in which the first action was carried out by a falling object (object condition). The second action in the sequence was always performed by the subject, during which they extracted the food reward made accessible in the first step. When this action sequence broke down, and the apparatus was not manipulated by the experimenter or the object such that the food reward could be extracted, subjects were faced with the problem of achieving the necessary end-state of the apparatus on their own. We probed subjects' responses to this breakdown across two phases: one in which they could access and act on the apparatus and one in which they could not. In the first phase, we investigated the hypothesis that chimpanzees in the partnered condition would more readily reproduce the experimenter's action to manipulate the apparatus, while those in the object condition would be less likely to independently produce the necessary first action in the sequence. Such a pattern of behaviour would show evidence for task co-representation – a scaffolding which is only available in the partnered (coordinated) condition of the learning phase. We also examined subjects' rate of communication towards the experimenter. We hypothesised that subjects in the partnered learning condition would communicate more frequently in an attempt to re-engage the experimenter's help (as in 48). We also investigated whether subjects were more likely to communicate before or after rendering the apparatus "impossible" by incorrectly performing their known second action in the sequence without having successfully performed the first action. If subjects were more likely to communicate prior to, rather than after, the incorrect action, it would suggest that they are able to track the roles in the sequence such that they are aware of when the experimenter's help would be of use to them.

In the second phase, subjects faced a similar breakdown in the action sequence, but were unable to access the apparatus to solve the task themselves. Given that, during the first phase, communication could have been suppressed in those subjects who successfully operated the apparatus, we used this second phase to explore communication behaviours in the absence of an ability to take independent action. As in the first phase, we hypothesised that subjects in the partnered learning condition would communicate more frequently than those who had never experienced collaboration with the experimenter. This effect, in both phases, might suggest that the partnered chimpanzees formed a specific expectation of collaborative action from the experimenter and therefore may have represented the task as coordinated, whereas the object chimpanzees had no such past experience.

## Methods

### Ethics

The subjects' enclosure included access to both indoor and outdoor areas, with food, water, and enrichment available ad libitum. Subjects were housed socially in a large (15 individuals), multi-male, multi-female group, mirroring the social structure of wild chimpanzee troops. Subjects were not separated from their group mates or closed off from their main enclosure during testing; subjects voluntarily approached the researcher's station within the Budongo Research Unit (BRU, see S1 Fig) in order to initiate a testing session, and could leave testing at any time. All research and husbandry complied with regulations set by the European Association of Zoos and Aquaria (EAZA) and the World Association of Zoos and Aquariums (WAZA). This method was approved in writing by the RZSS Edinburgh Zoo BRU Science Committee and by the University of St. Andrews School of Psychology and Neuroscience Research Ethics Committee.

### Subjects

Ten chimpanzees (*Pan troglodytes*), housed in the Budongo Research Unit (BRU) at RZSS Edinburgh Zoo, participated in this study. The subject group included 1 juvenile (aged 5 years) and 9 adults (aged 21–46 years).

### Apparatus

Each subject was tested by the same female experimenter. All subjects had a history of participation in interactive experiments with this same experimenter, including the provision of food. Testing was conducted in the research rooms of the BRU (see S1 Fig). The apparatus for this study consisted of a small wooden see-saw constructed with a ridge along one side (to prevent grapes from rolling off) and a shallow groove on the other side, into which a strip of paper was placed. The see-saw functioned such that if a grape was placed on the ridged side with the ridged side slanted down, and the see-saw was then flipped toward the grooved side, the grape would roll into the groove, where the subject could retrieve the food by pulling the strip of paper toward themselves (see Fig 1). The apparatus also included a weighted lever (constructed of a piece of wood with a weight attached near the top) affixed to a long piece of string. The lever rested on the table next to the grooved side of the see-saw, and the experimenter held the piece of string tightly under the table, out of view of the subject. The lever functioned such that, if the string was released, the lever would fall and hit the grooved side of the see-saw, flipping it toward that side, without the appearance that the experimenter had directly manipulated the apparatus (see Fig 2). See Supplementary Information for further details on the construction of the see-saw apparatus. A cardboard occluder was used to obscure the see-saw from view during baiting, so that subjects could not see the experimenter use her hands to orient the see-saw with the ridged side down at the start of each trial.

### Procedure

The experiment consisted of four phases, differentiated by subjects' access to the apparatus: a pre-training and habituation phase, a "blocked" learning phase, an "unblocked" test phase to probe for action-learning and communication

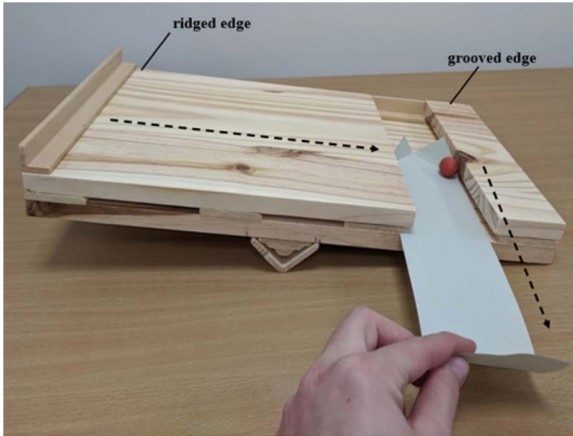

**Fig 1. Depiction of the see-saw apparatus flipped toward the grooved edge.** Dashed lines indicate the path of the grape as it rolls from the ridged edge to the grooved edge and is pulled toward the subject using the paper.

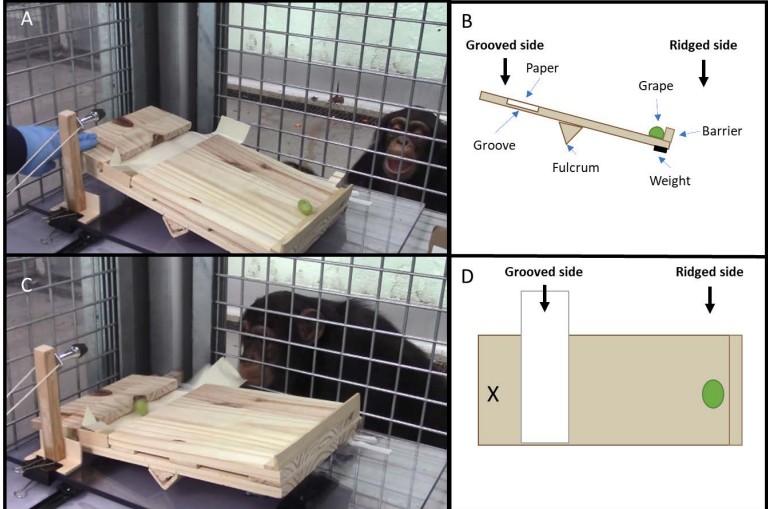

**Fig 2. Top left (A): See-saw apparatus in the starting position, about to be flipped by the experimenter.** Next to the experimenter's hand is the lever used in the object condition. Top right (B): graphic of see-saw apparatus at the start of a trial, before it has been flipped. Bottom left (C): After the see-saw has been flipped, the subject (Velu) is pulling the paper to retrieve the grape. Bottom right (D): overhead view of the see-saw apparatus at the start of a trial, before it has been flipped. X indicates the approximate location of the experimenter's hand or object (lever) when flipping the apparatus.

efforts, and a "re-blocked" test phase to probe for communication efforts alone. We used a between-subjects design, with subjects assigned to experimental conditions in a pseudo-randomized, counterbalanced manner, to ensure equal distribution of sex. See Table 1 for detailed subject demographics; the protocol was identical between groups during the pre-training, unblocked, and re-blocked phases, but different during the blocked learning phase. Subjects were divided into two groups – those who learned the apparatus with the experimenter flipping the see-saw ("partnered") and those who learned the apparatus with the lever falling onto the see-saw ("object"). The experimenter initiated trials only when the subject was seated within one arm's reach of the testing window and facing the apparatus. Across all phases, the experimenter did not proceed with trials if any subject in the opposite experimental condition was present

**Table 1. Subject demographic information and experimental condition assignment.**

| Name | Age | Sex | Experimental Condition |
|------|-----|-----|------------------------|
| Lucy | 43 | F | Partnered |
| Eva | 39 | F | Partnered |
| Qafzeh | 29 | M | Partnered |
| Frek | 26 | M | Partnered |
| Velu | 5 | M | Partnered |
| Kilimi | 27 | F | Object |
| Edith | 24 | F | Object |
| David | 46 | M | Object |
| Louis | 44 | M | Object |
| Liberius | 21 | M | Object |

in the testing space, and did not proceed with trials in the unblocked and re-blocked test phases if any other subject in either experimental condition was present, to prevent observational learning from conspecifics. Given that all chimpanzees had unobstructed access to the research space at all times, in order to avoid substantial delays to testing, the experimenter did proceed with trials in the blocked learning phase if another subject from the same experimental condition was in the room, as there was no concern that said subject might observe an alternative condition to the one they were assigned.

**Blocked learning phase.** Subjects first completed a series of habituation and pre-tests to ensure that they were comfortable using the paper to retrieve rewards from the apparatus and had the inhibitory control to wait until the grape had rolled onto the paper before pulling (see Supplementary Information and S1 Table for full details on this training).

The blocked learning phase constituted the between-subjects aspect of the experimental design. Subjects sat in front of the see-saw, facing the experimenter. The window to the left of the experimenter was blocked by a polycarbonate panel; subjects could not access the table from that side (see Fig 2. In both conditions, the see-saw began oriented toward the ridged side while the experimenter baited the groove with a piece of paper and placed a grape on the ridged side. This manipulation always occurred behind the occluder, so subjects could not observe the experimenter's actions. To initiate a trial, the experimenter pushed the table forward until the paper strip was within reach of the subject and removed the occluder. Then either:

*Partnered Condition (Coordination)*: While still facing toward the subject, the experimenter used her hand to flip the see-saw toward the grooved side.

*Object Condition (Non-Coordination)*: The experimenter turned her head and body to the right, away from the table, subject, and apparatus. While looking away, and out of view of the subject, she released the string of the weighted lever, such that it fell on the see-saw and caused it to flip toward the grooved side.

In both conditions, the grape then rolled down the see-saw, into the groove, and the subject was able to retrieve it by pulling the piece of paper. Subjects received 100 learning trials, grouped into 10-trial blocks, and subjects could receive up to two blocks per day. A video example of the protocol for this phase can be found at the following link: https://osf.io/sdj7w/?view_only=a8e6155134794723a5ad1e1d2481785b

**Unblocked phase.** In this phase, subjects had the opportunity to operate the see-saw themselves. This phase was intended to probe subjects' understanding of how to operate the see-saw, following their learning condition. This phase also examined whether the subjects produced any communicative behaviours toward the experimenter, in efforts to re-engage or recruit her help with the apparatus. Trials were initiated with the subject sitting in front of the see-saw, but the panel to the left of the experimenter was no longer blocked by polycarbonate; subjects could access the grooved end of the see-saw through the mesh on that side. The apparatus was baited and presented as in the blocked learning phase.

The experimenter then sat, with neutral posture and facial expression, facing the subject, and did not move to flip or interact with the apparatus, nor did she release the lever. Subjects were permitted to interact with the apparatus freely for 60 seconds; they had the opportunity to move around to the side and flip the see-saw themselves before pulling the paper. Subjects were also able to pull the paper out of the see-saw without flipping it and were able to manipulate the see-saw by other means (e.g., using a stick, using the paper as a lever, etc.). If the subject successfully acquired the grape through their manipulations of the see-saw, the trial ended immediately. This phase consisted of one 12-trial block per subject.

**Re-blocked phase.**  Prior to the re-blocked phase, subjects received one block of 10 trials using the same procedure as in the blocked learning phase, to refresh their memory of their original learning condition.

The re-blocked phase was intended to probe subjects' willingness or likelihood to produce communication behaviours toward the experimenter when she was no longer performing her role (for those who experienced the partnered learning condition) or when the apparatus was not functioning as expected (for those who experienced the object learning condition), and when the subjects were unable to operate the apparatus themselves. Trials proceeded as in the unblocked phase – the apparatus was baited and presented and the experimenter did not move to operate it, but the panel on the experimenter's left side was blocked once again, so subjects were unable to operate the see-saw themselves, even if they had innovated this solution in the unblocked phase. After a period of 10, 20, or 30 seconds, the experimenter used her hand to flip the see-saw and the subject was able to use the paper to retrieve the grape, if they had not already pulled it out of the apparatus. The experimenter was not responsive to any communicative behaviours from subjects – she maintained a neutral expression and position until the designated time to flip the see-saw. In the event that the subject managed to retrieve the grape with an alternative method, such as by using a stick, the trial was repeated (see S2 Table for a breakdown of repeated trials by subject). Subjects received two 12-trial blocks in this phase, for a total of 24 trials each.

**Note about disruptions to data collection.**  Testing in this experiment was interrupted by lockdown measures to prevent the spread of COVID-19. For details on modifications to the delivery of the protocol with respect to timing and interruptions, see S3 Table.

**Statistics and analysis.**  We coded subjects' actions toward the apparatus and communication behaviours toward the experimenter from video recordings using BORIS [49]. With regard to action, we scored the occurrence and latency of trial success (successful manipulation of the apparatus such that the subject acquired the grape). We further scored the method of success, categorised as either "flip" (in which the subject used their hand to press down the grooved side of the see-saw before pulling the paper), or "alternative" (in which the subject used any other means to orient the grooved side of the see-saw downward before pulling the paper, see Table 2). We also scored instances in which the subject first incorrectly pulled the paper, but subsequently manipulated the see-saw such that the grape rolled into the groove (that is, they performed both correct actions but in the wrong order). This response occurred only 2 times, both with the same subject, and so was excluded from analyses.

**Table 2.  Overview of success method and frequency amongst subjects with at least one trial success.**

| Subject | Condition | Method | Frequency |
|---|---|---|---|
| Velu | Partnered | flip | 6 |
| Frek | Partnered | flip | 6 |
| Eva | Partnered | alternative (paper lever) | 1 |
| Kilimi | Object | alternative (paper lever) | 3 |
| Edith | Object | alternative (paper lever) | 5 |
| David | Object | alternative (stick) | 1 |

We also scored occurrence and latency to (incorrectly) pull the paper before the see-saw was oriented toward the grooved side. This pull response was used to compute a pre-pull and post-pull state of the apparatus in each trial. Communication behaviours could therefore be categorised as "pre" or "post" pull according to their timing in the trial relative to the subject pulling the paper. On trials in which the subject did not pull at all, or pulled after correctly manipulating the see-saw, the pre-pull state lasted the entire length of the trial. Communication behaviours included a mixture of acts commonly used by captive chimpanzees to request food or express protest toward humans (see Table 3) and were included if they were performed within one arm's reach of the mesh panel. Communication behaviours were collapsed by type for each subject and analysed as the total count of communication behaviours per trial.

We analysed the effect of learning condition (partnered, object) on the occurrence and latency of subjects' actions by fitting generalised linear mixed models (GLMM, [50]). For occurrence of trial success (manipulating the see-saw to acquire the grape) and pulling (incorrectly pulling the paper without manipulating the see-saw), we conducted analyses using GLMMs with a binomial error structure and a logit link function. For method of trial success (flip, alternative), we conducted an analysis using a GLM with a binomial error structure and a logit link function. For latency of these actions, we conducted analyses using GLMMs with a Gamma error structure with a logarithmic link function. We analysed the effect of learning condition on the total count of communication behaviours per trial in each phase by fitting GLMMs with a Poisson error structure and logarithmic link function [51]. Using only those trials in which subjects produced at least one communication behaviour, we also fitted a Poisson GLMM to assess whether the state of the apparatus (pre- or post-pull) affected subjects' rate of communication.

For more information on the fixed and random factors included in these models, and for details concerning the process of fitting these models, see Supplementary Information, see also OSF: https://osf.io/sdj7w/?view_only=a8e6155134794723a5ad1e1d2481785b) Models were implemented using the *lmer* and *glmer* functions of R package *lme4* [52]. P-values for individual fixed effects and interaction terms were calculated using likelihood ratio tests of the fitted model with and without the relevant term (R function *drop1*, [53]). We calculated the effect size of condition and of the state of the apparatus for certain models, including odds ratios (for binomial models) and ratios of the estimated rate of communication (for the Poisson models) (R package *emmeans*, [54]). We assessed multicollinearity by calculating the generalised variance inflation factors of each model with random effects excluded, using the R package *car* [55]. There was no concern with multicollinearity in any models (GVIF<2 for all factors). All analysis was conducted in RStudio (version 4.2.1, [56]), and all data visualisations were created using the R package *ggplot2* [57].

**Reliability analysis.** We assessed interobserver reliability using Cohen's kappa for the occurrence of action behaviours and intraclass correlation coefficient for the latencies of actions and for the total count of communication behaviours per trial. We calculated ICC using two-way, absolute-agreement, random-effects models with 95% confidence intervals.

**Table 3. Ethogram of communication behaviours. See S4 Table for more detailed criteria for coding these behaviours.**

| Behaviour | Definition |
| --- | --- |
| tap | tapping or rapping fingers or hand against the table |
| finger thrust | thrusting fingers (or single finger) through the mesh |
| present mouth | presenting lips or mouth, open or closed, through the mesh |
| head nod | moving head up and down rapidly and repeatedly (at least two movements) |
| hand fling | raising hand upward and producing a flinging gesture toward the shoulder |
| pass paper | pushing or holding strips of paper, available from earlier trials, through the mesh |
| grumble | producing an audible, low-pitched vocalisation, akin to a grunt, grumble |
| whine | producing a high-pitched vocalisation akin to a whine or cry |
| raspberry | producing a lip buzzing/raspberry sound |

We divided communication behaviour coding between authors EW and ESM; each coder scored two thirds of each subject's trials. We found excellent reliability between these two coders using the one third of trials which they had both scored (unblocked phase: ICC = 0.93, CI = 0.9–0.96; re-blocked phase: ICC = 0.93, CI = 0.9–0.96). In the final dataset for each subject, we randomly selected either EW or ESM's scores from the overlapping data.

An additional coder, blind to the empirical questions and predictions of this study, scored actions and communication behaviours for 15% of trials to assess reliability of the coding scheme. Interobserver reliability was very good across all response variables (unblocked phase, trial success: K = 1, N = 18, $p < .001$; success method: K = 1, N = 18, $p < .001$; pulling: K = 1, N = 18, $p < .001$; latency to succeed (ICC = 0.99, CI = 0.97–1), and latency to pull: (ICC = 1, CI = 1–1); communication count: ICC = 0.85, CI = 0.66–0.94; re-blocked phase, communication count: ICC = 0.83, CI = 0.65–0.92).

## Results

### Action learning results (Unblocked Phase)

**Rate of success by any method.** A binomial GLMM found no effect of learning condition on subjects' likelihood to acquire the reward in the test phase by any method ($\chi2(1) = 0.23$, $p = 0.631$, $OR = 1.67$); collapsed across all methods, subjects were equally likely to successfully manipulate the apparatus in both the partner and object conditions. A Gamma GLMM found no effect of condition on action latency: $\chi2 (1) = 0.99$, $p = 0.284$. The learning condition did not impact the time it took subjects to manipulate the apparatus in the unblocked phase.

**Trial success method.** We conducted a binomial GLM across successful trials only, and found a significant effect of learning condition on the success method (flip vs alternative) ($\chi2(1) = 23.27$, $p < .001$, $OR = 2$), see Fig 3. This analysis included only subjects who succeeded in at least one trial (partnered condition: n = 3; object condition: n = 3), excluding those subjects who never succeeded (n = 4). Subjects who experienced the partnered learning condition were significantly more likely to flip the see-saw (as opposed to employing alternative methods) than those in the object learning condition. Notably, no chimpanzee in the object condition innovated the flip solution. There was no individual variation in method of success; subjects who manipulated the see-saw used exclusively one method or the other.

**Success consistency.** We conducted an exploratory binomial GLMM to examine whether consistency of success, like method of success, varied by condition. This analysis excluded those subjects who never succeeded. We found a significant interaction between condition and trial on the pattern of successes: subjects in the partnered learning condition tended to replicate their success on subsequent trials while subjects in the object condition did not consistently repeat

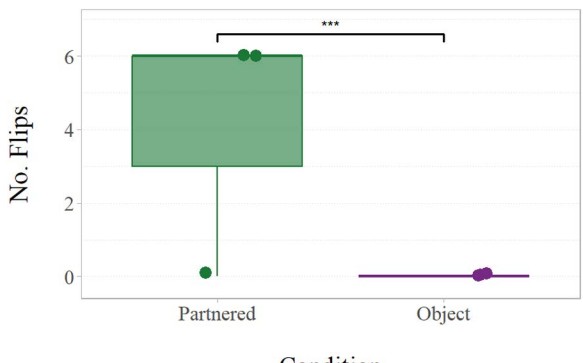

**Fig 3. Total number of flips (as opposed to alternative methods) by experimental condition (partnered, object) including only unblocked phase trials in which a success occurred (out of a possible 12).** Points depict individual subjects' data (n = 3 from each learning condition). The central line of the boxplot shows the median, the coloured boxes show the IQR, and the whiskers indicate the minimum and maximum values of the data within 1.5 x IQR. ***p < .001.

their success (Condition * Trial: $\chi2(1)$ = 3.7, $p=0.054$, *difference in estimated trends*$=-3.34$), see S2 Fig. That is, subjects in the object condition succeeded sporadically across trials, whereas subjects in the partnered condition tended to succeed in remaining trials following an initial success.

**Incorrect responses.** A binomial GLMM looking at incorrect pulls of the paper showed no effect of condition: $\chi2(1)$ = 0.2, $p=0.657$, *OR* = 0.7. The incorrect response of pulling the paper prior to manipulating the see-saw into a downward position did not vary significantly between the two conditions. The odds ratio showed that subjects in the object condition incorrectly pulled the paper 29.63% more often than subjects in the partnered condition. A Gamma GLMM including only incorrect trials showed no effect of condition on latency to incorrectly pull the paper: $\chi2(1)$ = 0.05, $p=0.821$.

## Communication results (Unblocked Phase)

**Overall rate of communication.** Subjects produced a mean (± SD) of 2.27 (± 2.03) communication behaviours per trial in the unblocked phase. A Poisson GLMM showed no significant effect of condition on the number of communication behaviours for a given trial ($\chi2(1)$ = 0.23, $p=0.63$, *ER*$=1.09$). There was a significant effect of trial success on communication behaviours; subjects produced double the amount of communication behaviours on trials where they did not ultimately succeed than on trials in which they succeeded ($\chi2(1)$ = 6.37, $p=0.012$, *ER*$=2$). There was no effect of trial number; subjects' rate of communication did not change across trials ($\chi2(1)$ = 0.68, $p=0.409$).

**State of the apparatus.** Communication in the unblocked phase varied significantly depending on the state of the apparatus (pre- or post-pull); subjects were more likely to communicate after incorrectly pulling the paper (at which point successful use of the apparatus becomes impossible) than before pulling the paper $\chi2(1)$ = 4.82, $p=0.028$, *ER*$=1.3$). Analysis of the ratio of estimated mean number of communication behaviours between states of the apparatus showed that subjects produced 29.95% more communication behaviours after they pulled the paper than before they had done so (see Fig 4).

## Communication results (Re-blocked Phase)

**Overall rate of communication.** Overall rate of communication: Subjects produced a mean (± SD) of 2.21 (± 2.35) communication behaviours per trial in the re-blocked phase. There was a non-significant trend of learning condition on the number of communication behaviours per trial; subjects who experienced the partnered learning condition produced marginally more communication behaviours than those who experienced the object learning condition ($\chi2(1)$ = 2.11,

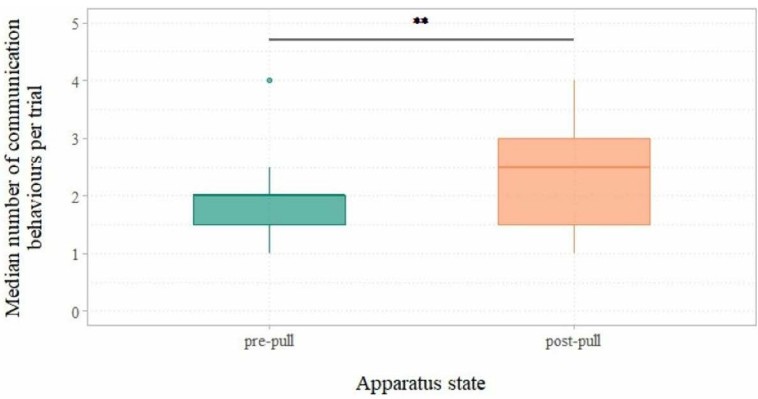

**Fig 4. Median number of communication behaviours per trial in the unblocked phase, separated by the state of the apparatus (pre- or post-pull).** The central line of the boxplot shows the median, the coloured boxes show the IQR, and the whiskers indicate the minimum and maximum values of the data within 1.5 x IQR. Data outside of this range are plotted as separate outlying points. **$p<.05$.

$p = 0.146$, $ER = 1.39$). Analysis of the ratio of estimated mean rates of communication between the two conditions showed that subjects in the partnered learning condition produced 38.57% more communication than those in the object learning condition. There was no significant effect of trial number on production of communication behaviours in the re-blocked phase ($\chi2(1) = 0.01$, $p = 0.941$).

## Discussion

This experiment investigated whether chimpanzees were able to reproduce a partner's actions after coordinating with them in a collaborative context, an effect which, if robust, would indicate task co-representation. We did not find evidence of action-learning facilitated by this coordination; there was no significant effect of learning condition (coordination vs non-coordination) on the rate of replicating the experimenter's actions, although the only two subjects who did produce this target action were in the coordinated learning condition. By this metric, we may conclude that chimpanzees did not categorically co-represent a coordination partner's actions, and the pattern of action-learning we did observe may have been a product of individual differences. During the course of this experiment we observed that certain subjects discovered alternative tactics to manipulate the apparatus: bootstrapping the end-state necessary for them to retrieve the reward without reproducing the action the experimenter had performed. The use of these alternative tactics was not evenly distributed between learning conditions; chimpanzees in the coordinated learning condition were significantly more likely to reproduce the experimenter's flipping action, whereas chimpanzees in the non-coordinated condition exclusively used alternative methods such as maneuvering the strip of paper as a lever to tilt the see-saw. These two methods of success differed in their fidelity to the experimenter's action; but they may also have differed in terms of action-planning. Those subjects who flipped the see-saw had to temporarily suspend their original action (pulling the paper), and move themselves to a different location, suggesting that these actions were goal-directed, rather than incidental effects of operating the apparatus in the rehearsed manner. Those subjects using the paper lever method, which accounted for all but one case of alternative success, were performing an action not incompatible with trying to execute their known (but now incorrect) action: pulling. Therefore, it is plausible that their only intention was to pull the paper, absent any action-planning regarding the movement of the apparatus.

These methods of success also varied in terms of subjects' performance across trials. The subjects who reproduced the experimenter's action, who were both in the coordinated condition, tended to do so consistently following their first success, while those using alternative methods tended to do so inconsistently, or only once, across their twelve trials. This interaction between trial and condition is further evidence for action-planning on the part of the subjects who flipped the apparatus, but not those who used the paper lever; flipping was a stable solution that subjects were able to execute in each trial, while alternative methods did not lend themselves to similar stabilisation. This difference in consistency of success between the subjects in the two learning conditions also cautions against fully accepting the null hypothesis regarding co-representation during coordination.

A rich interpretation of these differences in solution style, regarding fidelity to the experimenter's action, necessity of action-planning, and consistency of execution, is that chimpanzees in the coordinated condition co-represented their human partner's action, while those in the non-coordinated condition had no opportunity to do so. We may speculate that, if they engaged in coordination, subjects in the partnered condition represented their partner's actions as well as their own, which may have helped them consistently and intentionally reproduce that action themselves. Conversely, when the movement of the apparatus was caused by an object, subjects may only have learned about the affordances of the apparatus and later either emulated the necessary end-state of the see-saw using alternative means, or accidentally solved the apparatus in the course of executing their only known action. While our pattern of results is intriguing, we note that our sample size was insufficient to confidently reject an effect driven by individual differences in age, sex, manual dexterity, or motivation. We encourage replications of our design with other groups to investigate the robustness of the data obtained here, and to solidify our interpretations. Our sample size was further limited by the fact that only a subset (60%) of the

chimpanzees solved the task at all, which should be considered in future investigations and power analyses. While a simpler task may ensure a larger sample, our aim was to create a paradigm that would not be readily solved by all chimpanzees, to avoid ceiling effects and allow us to assess learning.

It is possible that the demonstration was more salient in the partnered condition than in the object condition. The experimenter's hand in the partnered condition may have directed subjects' attention and subsequent movement to the correct location on the apparatus via local enhancement [58] rather than via planned action following co-representation. Adult chimpanzees remember events performed by human hands better than by non-social mechanical claws, although the attention paid to hands (measured via looking time in eye-tracking studies) varies between experiments [59,60]. More research is needed, however, to further tease apart whether this pattern of results reflects an intentional consideration of a partner's actions, automatic social attentional processes, or perhaps something in between such as a system for understanding where to direct attention during cooperative activities.

The current study was designed to investigate whether a joint task can improve action learning, a finding that would suggest a facilitative impact of co-representation. The current paradigm resembles the structure of some social learning experiments (for example, [61]) in the sense that it investigates subjects' learning of an action previously performed by a social model. In our design, however, the socially modelled action is not just a demonstration, but is part of an interdependent sequence of actions between the subject and the experimenter. Our design offers a scaffolding of co-representation to action-learning, which is not necessarily mutually exclusive of social learning. It may be that co-representation and social learning rely on similar, or even related mechanisms; both require some representation of a social agent's actions. It is even possible that the interdependence of actions during a joint task could facilitate social learning (see 30 for a similar perspective on coordination), perhaps as an ecologically relevant scaffolding, given that many dyadic interactions between primates (for example, play-fighting) do offer the opportunity for action-learning (see [62,63,64] for related perspectives on primate play). In order to parse any difference between classic social learning and social learning scaffolded by coordination, future research should employ a control condition in which the actions are socially demonstrated but not interdependent. This control would establish whether chimpanzees learn actions more effectively from coordination than from social observation alone.

Our data support the hypothesis that chimpanzees will produce requesting or impatience behaviours during a breakdown in collaboration (as in [65–68] and [48], but contrary to [45] and [69]). The breakdown in collaboration reliably solicited communication from all subjects in both phases, although this effect did not vary between conditions. For all subjects, regardless of learning condition, the experimental procedure created a collaborative sequence of actions in which the experimenter willingly, and without prompting, prepared, baited, and presented the apparatus. Although only the subjects in the partnered condition experienced the final two actions in the sequence as jointly dependent, both groups could have represented each trial as a collaborative endeavour in which the experimenter played a relevant role in their acquisition of the food reward. Additionally, it is possible that chimpanzees were aware that the experimenter was covertly manipulating the lever in the object condition. In response to a breakdown in this collaboration, subjects in both learning conditions produced communication behaviours toward the experimenter when the expected sequence of actions was not completed. This finding suggests that chimpanzees were using their gestures to solicit action from the experimenter, especially given our finding that subjects produced significantly fewer gestures in trials in which they succeeded at manipulating the apparatus. This paradigm may have been particularly suited to soliciting such a communication response; the subjects were reacting not only to a need for collaborative action from a partner, but also to the violation of the established expectation of help from the experimenter. This is in keeping with existing literature, where chimpanzees have not been shown to spontaneously communicate to recruit or reengage a partner in a coordinated activity with no preexisting precedent [45,70], but did show a higher frequency of communication behaviours in task which involved a similar breakdown of established coordination (for example, [48]). The current finding is also consistent with work by Heesen et al. [71,72], which finds that chimpanzees and bonobos will produce communication behaviors during the breakdown of a dyadic activity between

conspecifics, such as grooming, suggesting that great apes may use communication more broadly to coordinate and sustain joint interactions.

We found evidence to suggest that the communication in this experiment was provoked by the subjects' perceived or actual inability to act on the apparatus. That is, chimpanzees communicated predominantly when they had already exhausted the possible actions they could take alone. In the unblocked phase, subjects produced a significantly higher rate of communication in trials in which they were ultimately unsuccessful – trials in which they did not successfully flip the see-saw. Notably, this effect emerged even when the amount of time available in which to communicate was controlled in the model; the difference in communication is not explained by the fact that proficient flippers tended to manipulate the see-saw very early in the trial, leaving little time to communicate. This evidence suggests that subjects' communication was provoked by their perceived inability to act on the apparatus. Although it was always possible for them to manipulate the see-saw in the test phase, if they were not aware of any action they could take to do so, then they may have resorted to expressing frustration or soliciting help from the experimenter. The effect is corroborated by the finding that, in the unblocked phase, subjects communicated more frequently after having (incorrectly) pulled the strip of paper than before having done so. The same explanation applies: while there is still an action, any action, that the subject can take on the see-saw independently, even if they understand that it is not the correct sequence of events, they are more likely to take that action than to communicate, and begin communicating only after they believe they have exhausted all of their own possible actions.

This finding is in line with existing literature on collaborative tasks showing that apes tend to act alone, rather than with a partner, when possible [43,73]. Similarly, [41] found that chimpanzees did not solicit conspecific partners to assist them in a joint task, but did solicit human partners once they had learned that the task could not be solved individually; chimpanzees do not resort to recruitment of a partner as their first port-of-call. The current finding is also consistent with apes' patterns of behaviour in stag-hunt paradigms, where chimpanzees have been shown to coordinate to acquire a high-value target, discarding a low-value reward that can be acquired individually, but they do not resort to communication to coordinate their behaviour and are most successful at coordinating when they can easily observe the choices of their partner [44,74]. The above literature suggests that chimpanzees' capacity to act individually is a relevant factor in whether they will solicit help using communication, since apes do communicate with cooperative human partners in some situations. For example orangutans spontaneously gestured to request food rewards from human partners in a task in which they were not able to produce any direct actions toward the reward themselves, and chimpanzees have been shown to use communication to sustain coordination, but only in situations where they had no actions available to them and could only proceed if the coordinating partner performed their role [48,67].

The present study focuses on interactions between chimpanzees and human experimenters, and thus the finding that chimpanzees communicate in a collaboration breakdown may not generalise to conspecific interactions. Existing literature has shown that ape subjects are more likely to communicate with and solicit help from a human experimenter than from a conspecific in collaborative tasks [37,41,75]. Relatedly, apes can develop and act on expectations that human experimenters will provide food [76], but may not get similar exposure to helpfulness from conspecifics, who have been shown to cease or withhold help unless they will also experience personal gain ([77,78], but see [79,80] for contradictory findings). Captive apes' social relationships with humans may therefore evoke different expectations of helping behaviour than their relationships with conspecifics, such that human caregivers and experimenters may be more readily seen as reliable social tools (see 39 for a description of social tool use in apes). Whilst chimpanzees may be less likely to communicate with conspecific partners in collaboration breakdowns, this is not evidence of a lack of ability to do so; chimpanzees' rates of communication may be limited by their perception of their partner's willingness to help. The finding that apes communicated in a collaboration breakdown in the context of our experiment contributes to a fuller picture of the proximal factors which may support communication during collaborative tasks: sufficient expectation that the task partner is willing and able to help, sufficient motivation to complete the task on the part of the communicating ape, and a task structure that allows for clear, easy communication. While the interspecific nature of the interaction in this task could be seen as a

limitation to the ecological validity of this experiment, it may also be seen as evidence of a broader principle about chimpanzee communication.

Here, we have presented a design which measured the facilitatory effect of co-representation on action learning during a coordination breakdown, and explored communication behaviour as a measure of expected coordination from a partner. This study highlights the importance of qualitative differences in action reproduction, which may reveal patterns of solution that vary by condition, even when the overall count of successful trials does not. In order to investigate the effects of co-representation, it is crucial to consider style and consistency of actions, as well as occurrences. Although the small sample of subjects who solved this task limits our conclusions with regard to apes' capacity for co-representation, the patterns of behaviour we observed reveal several key considerations for future study of the cognitive processes of collaboration. This study suggests that communication may be a useful metric to evaluate apes' representation of the collaborative nature of a task, given the appropriate task conditions. Namely, rather than a choice between action or communication, we recommend a trial structure in which the subjects are provided time to act on the apparatus and also communicate, including situations where they may choose to communicate only after exhausting all possible solo actions. While apes may not be motivated to use communication to engage coordination as a first port-of-call, this is not evidence of a failure to represent the task as a collaborative one. Communication after failed solo actions may still reflect an expectation of action from a partner on the basis of pre-established coordination. We also suggest that future investigations consider the wider context of the task such as past history of coordination on similar or other tasks, and the relevance of the collaborator to their success, including aspects such as the baiting and presentation of apparatuses prior to any critical condition differences. Primate subjects may take such factors into account when judging a task as collaborative or not, which may affect their expectations of collaboration from a human experimenter. For communication behaviour to reflect apes' expectation of action, any differences between conditions must impose actual differences in the partner's relevance to the subjects' receipt of a reward.

When a previously joint task becomes a solo one, do apes learn from the experience of having collaborated? We propose that this question, and its bearing on the capacity for co-representation, is an area worthy of deeper investigation. Co-representation, and more broadly the ability to intentionally coordinate, fit into a larger puzzle regarding apes' social cognition and the evolutionary path to human collaboration. It is critical to develop tasks which can parse the nuances of individuals' cognitive engagement with collaborative endeavours. While the current study does not give a conclusive answer to the question of co-representation in apes, the observed pattern of results is promising, and we highlight several considerations to more effectively probe and measure it in future work.

## Supporting information

**S1 Figure. Bird's-eye-view image of the Budongo Research Unit (BRU) with relevant research areas and research panels labeled.** "Research Area" indicates areas where human researchers can safely sit or stand to conduct experimental research with the chimpanzees at Edinburgh Zoo. All research areas are separated from chimp areas by metal walls or by mesh/polycarbonate panels (numbered 1–11 below). "Chimp Area" indicates areas where chimpanzees can enter or exit freely during research times. These areas are accessed via one of the chimpanzee's indoor enclosure pods (Chimp Pod No. 1), and are divided by hydraulic doors (H1, H2, and H3), which are always open during research times. This experiment was conducted in the alcove contained by windows 7, 8, and 9.
(TIF)

**S2 Figure. Consistency of success across trials, separated by success method and learning condition, and faceted by individual.** Subjects in the partnered learning condition, specifically those who used the "flip" method of success, tended to replicate their success consistently over subsequent trials, while those in the object condition, using alternative methods, succeeded only sporadically across trials.
(TIF)

**S1 Table. Table of the number of sessions administered to each subject in pre-test 2, before they reached minimum passage criteria (13/16 correct critical trials across two consecutive blocks of 8 critical trials each; trials with a 0-second delay were not included in this metric).** All subjects received one initial session of pre-test 2 before attempting to meet passage criteria, thus the minimum possible number of sessions is 3.
(DOCX)

**S2 Table. Number of trials repeated in the re-blocked phase due to the subjects' use of alternative methods to flip the seesaw.**
(DOCX)

**S3 Table. Table depicting the timing of each subject's progress through the experiment, with lapses in testing included.** The phase or trials listed for each subject in each time period indicate the starting (S) and final (F) piece of testing completed within that time period. Any phase that was incomplete and restarted in the next period following a delay is indicated with an (x).
(DOCX)

**S4 Table. Detailed ethogram for identification and discrimination of communication behaviours; this exact ethogram was provided during training for reliability coding.**
(DOCX)

## Acknowledgments

We thank Eloise Dallas for conducting the reliability coding for these data, and Kate Grounds for her role as research coordinator at Edinburgh Zoo. We also thank the workshop team in the University of St Andrews School of Psychology and Neuroscience for their support with apparatus construction. We are grateful to the Royal Zoological Society of Scotland (RZSS) keeping and veterinary staff for their care of animals and technical support throughout this project. We thank Zoë Goldsborough and three other anonymous reviewers for their helpful feedback.

## Author contributions

**Conceptualization:** Elizabeth Warren, Emma Suvi McEwen, Josep Call.

**Data curation:** Elizabeth Warren, Emma Suvi McEwen.

**Formal analysis:** Elizabeth Warren, Emma Suvi McEwen, Josep Call.

**Funding acquisition:** Josep Call.

**Investigation:** Elizabeth Warren, Emma Suvi McEwen.

**Methodology:** Elizabeth Warren, Emma Suvi McEwen, Josep Call.

**Project administration:** Elizabeth Warren.

**Resources:** Elizabeth Warren, Emma Suvi McEwen, Josep Call.

**Supervision:** Josep Call.

**Validation:** Elizabeth Warren, Emma Suvi McEwen.

**Visualization:** Elizabeth Warren, Emma Suvi McEwen.

**Writing – original draft:** Elizabeth Warren, Emma Suvi McEwen.

**Writing – review & editing:** Elizabeth Warren, Emma Suvi McEwen, Josep Call.

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
