## [Decision Letter · Decision Letter 0]

5 Mar 2025

PONE-D-25-05158Do chimpanzees (Pan troglodytes) mentally represent collaboration?: Action-learning and communication in a partnered taskPLOS ONE?

Dear Dr. McEwen,

Thank you for submitting your manuscript to PLOS ONE. After careful consideration, we feel that it has merit but does not fully meet PLOS ONE’s publication criteria as it currently stands. Therefore, we invite you to submit a revised version of the manuscript that addresses the points raised during the review process.

We look forward to receiving your revised manuscript.

Kind regards,

Miquel Llorente, PhD

Academic Editor

PLOS ONE

Journal Requirements:

This work was supported by the European Union’s Seventh Q20 Framework Programme (FP7/2007–2013)/European Research Council Grant 609819 (SOMICS)

We thank Eloise Dallas for conducting the reliability coding for these data, and Kate Grounds for her role as research coordinator at Edinburgh Zoo. We also thank the workshop team in the University of St Andrews School of Psychology and Neuroscience for their support with apparatus construction. We are grateful to the Royal Zoological Society of Scotland (RZSS) and the University of St Andrews for core financial support to the Edinburgh Zoo’s Budongo Research Unit where this project was carried out. We are grateful to the RZSS keeping and veterinary staff for their care of animals and technical 37 support throughout this project. This work was supported by the European Union’s Seventh Q20 Framework Programme (FP7/2007–2013)/European Research Council Grant 609819 (SOMICS). Edinburgh Zoo’s Budongo Research Unit is core supported by the Royal Zoological Society of Scotland (Registered charity number: SC004064) through funding generated by its visitors, members and supporters, and by the University of St Andrews (Registered charity number: SC013532) who core supports the maintenance and management costs of the research facility

This work was supported by the European Union’s Seventh Q20 Framework Programme (FP7/2007–2013)/European Research Council Grant 609819 (SOMICS)

5. http://www.bmj.com/content/340/bmj.c181.long. Please remove all personal information, ensure that the data shared are in accordance with participant consent, and re-upload a fully anonymized data set.

6. Please%20include%20captions%20for%20your%20Supporting%20Information%20files%20at%20the%20end%20of%20your%20manuscript,%20and%20update%20any%20in-text%20citations%20to%20match%20accordingly.%20Please%20see%20our%20Supporting%20Information%20guidelines%20for%20more%20information:%20http:/journals.plos.org/plosone/s/supporting-information.%20" Please include captions for your Supporting Information files at the end of your manuscript, and update any in-text citations to match accordingly. Please see our Supporting Information guidelines for more information: http://journals.plos.org/plosone/s/supporting-information.

Additional Editor Comments:

Dear authors,

Thank you for submitting your manuscript to PLOS ONE. I have now received and carefully considered the reviews from two independent reviewers. Based on their assessments, I am pleased to inform you that your manuscript is generally well-received and offers a valuable contribution to the study of co-representation and problem-solving in captive chimpanzees. However, before it can be accepted for publication, minor revisions are required.

Reviewer 2 raised concerns about the sample size, noting that while small sample sizes are common in primatology, the limited number of successful participants in your study makes it difficult to draw strong conclusions. The reviewer acknowledges the methodological soundness of your approach but encourages additional discussion on the limitations imposed by the sample size and the potential implications for the robustness of your findings. While further data collection would be ideal, it is not a mandatory requirement at this stage; instead, clarifying the limitations in the discussion would be beneficial.

Reviewer 1 provided more specific recommendations regarding the structure and clarity of the manuscript. They highlighted the importance of introducing the distinction between cross-species vs. within-species collaboration earlier in the paper, particularly in the introduction. While this point is well-addressed in the discussion, the reviewer believes that a more explicit rationale at the beginning would improve the manuscript’s coherence and provide clearer context for the study’s design and findings.

Additionally, Reviewer 2 noted some redundancies in the introduction and suggested streamlining certain sections to enhance clarity and conciseness. They also raised a methodological consideration regarding the object condition, questioning whether the mere presence of a human observer might have influenced chimpanzees’ perceptions of the task. Furthermore, they suggested discussing the possibility of local enhancement as an alternative explanation for the observed behavior in the partner condition.

To address these concerns, please provide the following revisions:

- Clarify the limitations imposed by the sample size in the discussion.

- Introduce the issue of cross-species collaboration earlier in the manuscript, providing a clearer rationale for the study’s design.

- Streamline the introduction to reduce repetition and enhance clarity.

- Discuss the potential influence of human presence in the object condition.

- Consider the role of local enhancement as an alternative explanation in the discussion.

In addition, please carefully review the specific line-by-line comments provided by Reviewer 1 and 2 to address clarifications and edits.

I believe these revisions will significantly strengthen your manuscript and improve its clarity and impact. Please submit your revised version along with a point-by-point response detailing how each comment has been addressed.

I look forward to receiving your revised manuscript.

Best regards,

Dr. Miquel Llorente

Reviewers' comments:

Reviewer's Responses to Questions

**Comments to the Author**

1. Is the manuscript technically sound, and do the data support the conclusions?

Reviewer #1: Yes

Reviewer #2: Partly

2. Has the statistical analysis been performed appropriately and rigorously?

Reviewer #1: Yes

Reviewer #2: Yes

3. Have the authors made all data underlying the findings in their manuscript fully available?

Reviewer #1: Yes

Reviewer #2: Yes

4. Is the manuscript presented in an intelligible fashion and written in standard English?

Reviewer #1: Yes

Reviewer #2: Yes

Reviewer #1: This study is a well-thought out and thoroughly executed assessment of mechanisms underlying collaboration in captive chimpanzees. Through a two-step task, either with help of a human partner or a subject, the authors consider whether chimpanzees form some mental representation of the actions of their partner, and if they can integrate this information in their own actions. Using a between-subjects design, the authors tested chimpanzees’ performance on the task as well as their attempts to communicate with the human experimenter. They found that individuals in both the object and partner condition could successfully reproduce the necessary action, however, individuals in the partner condition were more likely to replicate the actions of the experimenter, while individuals in the object condition were more likely to find an alternative solution. Communication mostly occurred when chimpanzees had exhausted the possible actions they could take. The article is well-written, and I especially commend the good use of figures and supplementary material which make it very easy to follow the experimental set-up.

I appreciate the rigor and detail that went into planning the methodology and set-up of this study and in its execution. My main qualm with the manuscript in its current form does not lie in the content, but in the order in which it is presented, and I have a few minor comments.

Cross-species vs within-species collaboration

The authors acknowledge this point in the discussion, which I thought was a very well-presented consideration of this limitation, but I really missed this information earlier in the manuscript. The study presented here considers whether chimpanzees are capable of co-representing the mental state of a human partner, not a conspecific. It is very well possible that the mechanisms that underlie collaboration and mental perspective-taking cross-species are not entirely the same ones that underlie collaboration within-species. I understand the limitations of performing such experiments that led to the use of a human partner, but I would appreciate more discussion of this fact already in the introduction of the paper.

Can it truly be assumed that the cognitive capacities are the same for cross-species and within-species? In the rationale of the introduction, the authors explain how collaboration between individuals is crucial to survival of group-living species. Is collaboration with other species just as crucial? Or is it expected that if this capacity exists within species, it translates to between species without any issues? I think the choice of the authors to take a chimpanzee-human collaboration as analogue to a chimpanzee-chimpanzee collaboration merits some background in the introduction already, and some justification why between-species collaboration could be used as a proxy for assessing collaboration capacity within-species.

In the discussion, the authors actually refer to studies (629-633) which support the idea that chimpanzees represent collaboration with a conspecific and a human partner differently. Chimpanzees are actually more likely to collaborate with human partners. To me, this is very relevant information that should be discussed in the rationale and set-up of the experiment. In my view, these studies, and the finding of this manuscript, could reflect that captive chimpanzees specifically have become used to seeing humans as a source of food. Conspecifics are not, so they are less motivated to collaborate with conspecifics, since, unlike with humans, they are not sure a food reward would follow. To me this further signals that there might be a different process at work in cross-species collaboration than within-species, which certainly merits more discussion at earlier places in the manuscript (though I do think this paragraph in the discussion was excellent and very informative!).

Repetition/Lengthiness

Overall, I found the introduction quite lengthy, and felt as if it was at points repetitive. Especially the notion that primates’ ability for shared representations is not well understood was repeated several times. For instance, the paragraph from 142-157 could be rewritten to be much clearer and more concise. Now the same information is repeated for collaborative experiments and stag-hunt, when the relevant information from these studies is very similar. Also, this is likely a copy-paste error, but the paragraph in lines 536-549 is repeated in lines 550-563.

Object vs partner

After reading the manuscript and watching the supplementary material, I wonder to what extent the chimpanzees truly experienced the object condition as being devoid of human intervention. I understand why the current set-up was chosen, and I think it is a clever set up. However, I am not entirely sure if the mere presence of the human observer does not lead to the chimpanzee assuming the human had some kind of role in the flipping of the see-saw, even if they don’t see the exact mechanism. After all, in a zoo setting, chimpanzees are well-acquainted with humans making things happen of which they do not fully understand the mechanism and the causality is not clear (e.g., a caretaker pressing a button to open a hatch far away), so in that sense, I would not be surprised if in both conditions the chimpanzee assumes the human did something to flip the see-saw.

The difference in performance between the object and partner condition, with chimpanzees being more likely to copy the experimenter’s action in the partner condition, could possibly also be attributed to local enhancement. The chimpanzees observe the human experimenter touching the side of the see-saw, and as a result, are more likely to interact with this specific location themselves, resulting in the flip. This alternative would be less cognitively complex than chimpanzees representing their human partner’s mental state, and I think it warrants some mention in the discussion section.

Specific comments line by line

15: So here I would add “cross-species” in front of co-representation, to be very clear.

79: Is the partner in the Joint Simon studies presented here a conspecific or a heterospecific?

225: In what context was the female experimenter familiar? This warrants some more information. Did the subjects have experience with the experimenter providing food? Or was she a neutral observer to them? If food was given in the past, had all chimpanzees experienced this?

271-274: This section was vague to me, and I do think it is important information being shared here that now does not come across. If I understood correctly, trials in the blocked condition did continue if a conspecific of the same experimental condition was present, right? I would re-word this to be a) entirely clear in which situations trials continued with others present b) include justification why you made the decision to continue or interrupt some trials with an audience but others not, and c) in the results, I’d expect to see some information in how many trials conspecifics were present in the trial types where you did allow this. I think you probably have good reasons for deciding on this protocol, but as a reader it is not clear on what grounds these decisions were made, when I think this is relevant information.

446: how many subjects never succeeded?

516: There’s a punctuation mark too many behind “see-saw”.

609-628: I thought this was a very interesting discussion and good interpretation of the authors’ findings!

651: I think it is fascinating that we find this difference, to me it indicates that chimpanzees might be very intelligent in the sense that they are aware of when their efforts would be wasted vs when they are likely to get a food reward. It’s another reason why I think it is important to expand on the previous experience the chimpanzees in this study had with the experimenter specifically, since if they already associated her with food rewards they might be more inclined to cooperate/communicate.

Reviewer #2: The manuscript describes a study investigating the possibility for co-representation in captive chimpanzees collaborating with a human partner or an automatic mechanism in a simple food acquisition task. The manuscript is generally well-written and describes a novel approach to the problem of identifying co-representation in this context. As a minor concern, the style of writing could be simplified to make the content more accessible to a broader audience. This is an important consideration given the scope of the journal. Specific corrections are provided in the feedback.

The methodology appears to be sound and seemingly addresses the question that is being posed. The only major issue evident in this study is the limited sample size, complicated by the fact that only six of the ten individuals successfully completed the task. The small sample makes it difficult to discern what the responses of the animals mean but also whether the test itself is as appropriate in addressing the question posed as it seems. It also complicates the interpretation of potential causal factors in addition to those already mentioned which may have driven the responses of the subjects (e.g. differences in cognitive ability, age/sex differences). To their credit, the authors do identify the sample size as a problem in the study but, as a reviewer, it is difficult to look past this limitation in order to determine the scientific implications of the findings. I would urge the authors to increase their sample size in order to improve the confidence that the reader can place in the outcomes and improve the scientific impact that the study could have.

**Do you want your identity to be public for this peer review?** For information about this choice, including consent withdrawal, please see our Privacy Policy

Reviewer #1: **Yes: ** Zoë Goldsborough

Reviewer #2: No

---

## [Author Response · Author response to Decision Letter 0]

22 Apr 2025

Response to Reviewers

Editor:

1. Role of funder statement:

This work was supported by the European Union’s Seventh Q20 Framework Programme (FP7/2007–2013)/European Research Council Grant 609819 (SOMICS). Edinburgh Zoo’s Budongo Research Unit is core supported by the Royal Zoological Society of Scotland (Registered charity number: SC004064) through funding generated by its visitors, members and supporters, and by the University of St Andrews (Registered charity number: SC013532) who core supports the maintenance and management costs of the research facility. The funders had no role in study design, data collection and analysis, decision to publish, or preparation of the manuscript.

2. Data anonymity:

We acknowledge that the editor has asked us to remove any identifying information from human subjects and provide an anonymized data set. We reiterate here, as in the manuscript, that our experiment was conducted with non-human primates only, and thus the existing dataset contains no protected or identifying information for any human subjects.

3. Ethics statement:

We have added a full ethics statement, including the full name of the ethics committee that provided written consent, to the methods. It is also reproduced below:

Lines 223-235: The subjects’ enclosure included access to both indoor and outdoor areas, with food, water, and enrichment available ad libitum. Subjects were housed socially in a large (15 individuals), multi-male, multi-female group, mirroring the social structure of wild chimpanzee troops. Subjects were not separated from their group mates or closed off from their main enclosure during testing; subjects voluntarily approached the researcher’s station within the Budongo Research Unit (BRU, see Supplementary Materials Figure 1) in order to initiate a testing session, and could leave testing at any time. All research and husbandry complied with regulations set by the European Association of Zoos and Aquaria (EAZA) and the World Association of Zoos and Aquariums (WAZA). This method was approved in writing by the RZSS Edinburgh Zoo BRU Science Committee and by the University of St. Andrews School of Psychology and Neuroscience Research Ethics Committee.

Reviewer 1:

1. Introduce the issue of cross-species collaboration earlier in the manuscript, providing a clearer rationale for the study’s design.

We have specified earlier in the manuscript that the task involved is interspecific coordination, and added the clarified cation that this study was intended to investigate a species capacity for co-representation in general at all, rather than a specific tendency to engage in co-representation with conspecifics. Given that multiple studies have shown that human-given social cues can affect chimpanzees’ responses in communication, social learning and theory of mind studies (e.g., Hare et al., 2006; Haun & Call, 2008; Herrmann & Tomasello, 2005; see Call & Tomasello, 2024 for a review), we think that our use of cross-species collaboration is warranted.

Line 55-58: Here, we investigate chimpanzees’ action learning and communication behaviours during an interspecific collaborative task, to explore chimpanzees’ the species capacity for co-representation and the cognitive mechanisms at play in their collective behaviour.

2. Streamline the introduction to reduce repetition and enhance clarity.

We have removed several summarizing sentences which consolidate the existing literature, in order to streamline the introduction. See removed sentences at lines 73, 147, and 157.

3. Clarify context in which the experimenter was familiar to subjects

Each subject in this study had participated in previous interactive experiments with the same experimenter, and so had experience with that experimenter providing food and access to apparatuses in a research setting. This has now been clarified:

Line 243-245 “Each subject was tested by the same female experimenter. All subjects had a history of participation in interactive experiments with this same experimenter, including the provision of food.”

4. Clarify methodological details regarding the presence of other chimps during testing

We have clarified that trials in the blocked learning phase did proceed if subjects in the same experimental condition were also in the room, and we have also clarified the reason for this decision: chimpanzees had unfettered access to the research space at all times, and it would have delayed the 100 learning trials across a minimum ten separate sessions substantially if testing could not proceed when any of nine individual chimpanzees chose to be in the room. It was not a concern to us to count trials where the testing individual may or may not have been observed by non-interfering conspecifics, as we did not expect this to impact their learning, and it was not possible to collect the number of trials that individuals may have observed, other than their own trials, as it was possible for non-testing subjects to come in and out repeatedly during a session. We also did not run trials if any chimpanzees present were close enough to interfere with or displace the testing individual; other chimpanzees in the room were normally engaged in other activities such as play, sleep, or interacting with the zookeepers.

Line 294 -298 Given that all chimpanzees had unobstructed access to the research space at all times, in order to avoid substantial delays to testing, the experimenter did proceed with trials in the blocked learning phase if another subject from the same experimental condition was in the room, as there was no concern that said subject might observe an alternative strategy to the one they were assigned.

5. Clarify the number of subjects who never succeeded

We have clarified that 4 subjects never succeeded:

Line 460-461 “excluding those subjects who never succeeded (n=4).”

6. There’s a punctuation mark too many behind “see-saw”.

We have corrected it - thank you for catching this!

7. Discuss the potential influence of human presence in the object condition.

We have added this point to our discussion about the object condition possibly being perceived as cooperative by the chimpanzees.

Line 637: “Additionally, it is possible that chimpanzees were aware that the experimenter was covertly manipulating the lever in the object condition.”

8. Discuss the role of local enhancement as an alternative explanation in the discussion.

We have added the following a paragraph to the discussion to address this point.

Line 583: “It is possible that the demonstration was more salient in the partnered condition than in the object condition. The experimenter’s hand in the partnered condition may have directed subjects' attention and subsequent movement to the correct location on the apparatus via local enhancement (Thorpe, 1963) rather than via planned action following co-representation. Adult chimpanzees remember events performed by human hands better than by non-social mechanical claws, although the attention paid to hands (measured via looking time in eye-tracking studies) varies between experiments (Howard et al., 2017; Padberg et al., 2025). More research is needed, however, to further tease apart whether this pattern of results reflects an intentional consideration of a partner’s actions, automatic social attentional processes, or perhaps something in between such as a system for understanding where to direct attention during cooperative activities.“

Reviewer 2:

1. Clarify the limitations imposed by the sample size in the discussion.

We have expanded our discussion of the limitations:

Line 573 “While our pattern of results is intriguing, we note that our sample size was insufficient to confidently reject an effect driven by individual differences in age, sex, manual dexterity, or motivation. We encourage replications of our design with other groups to investigate the robustness of the data obtained here, and to solidify our interpretations. Our sample size was further limited by the fact that only a subset (60%) of the chimpanzees solved the task at all, which should be considered in future investigations and power analyses. While a simpler task may ensure a larger sample, our aim was to create a paradigm that would not be readily solved by all chimpanzees, to avoid ceiling effects and allow us to assess learning.”

Line-by-line comments:

Reviewer #1: This study is a well-thought out and thoroughly executed assessment of mechanisms underlying collaboration in captive chimpanzees. Through a two-step task, either with help of a human partner or a subject, the authors consider whether chimpanzees form some mental representation of the actions of their partner, and if they can integrate this information in their own actions. Using a between-subjects design, the authors tested chimpanzees’ performance on the task as well as their attempts to communicate with the human experimenter. They found that individuals in both the object and partner condition could successfully reproduce the necessary action, however, individuals in the partner condition were more likely to replicate the actions of the experimenter, while individuals in the object condition were more likely to find an alternative solution. Communication mostly occurred when chimpanzees had exhausted the possible actions they could take. The article is well-written, and I especially commend the good use of figures and supplementary material which make it very easy to follow the experimental set-up.

I appreciate the rigor and detail that went into planning the methodology and set-up of this study and in its execution. My main qualm with the manuscript in its current form does not lie in the content, but in the order in which it is presented, and I have a few minor comments.

Cross-species vs within-species collaboration

The authors acknowledge this point in the discussion, which I thought was a very well-presented consideration of this limitation, but I really missed this information earlier in the manuscript. The study presented here considers whether chimpanzees are capable of co-representing the mental state of a human partner, not a conspecific. It is very well possible that the mechanisms that underlie collaboration and mental perspective-taking cross-species are not entirely the same ones that underlie collaboration within-species. I understand the limitations of performing such experiments that led to the use of a human partner, but I would appreciate more discussion of this fact already in the introduction of the paper.

Can it truly be assumed that the cognitive capacities are the same for cross-species and within-species? In the rationale of the introduction, the authors explain how collaboration between individuals is crucial to survival of group-living species. Is collaboration with other species just as crucial? Or is it expected that if this capacity exists within species, it translates to between species without any issues? I think the choice of the authors to take a chimpanzee-human collaboration as analogue to a chimpanzee-chimpanzee collaboration merits some background in the introduction already, and some justification why between-species collaboration could be used as a proxy for assessing collaboration capacity within-species.

In the discussion, the authors actually refer to studies (629-633) which support the idea that chimpanzees represent collaboration with a conspecific and a human partner differently. Chimpanzees are actually more likely to collaborate with human partners. To me, this is very relevant information that should be discussed in the rationale and set-up of the experiment. In my view, these studies, and the finding of this manuscript, could reflect that captive chimpanzees specifically have become used to seeing humans as a source of food. Conspecifics are not, so they are less motivated to collaborate with conspecifics, since, unlike with humans, they are not sure a food reward would follow. To me this further signals that there might be a different process at work in cross-species collaboration than within-species, which certainly merits more discussion at earlier places in the manuscript (though I do think this paragraph in the discussion was excellent and very informative!).

See above: Reviewer 1, point 1

Repetition/Lengthiness

Overall, I found the introduction quite lengthy, and felt as if it was at points repetitive. Especially the notion that primates’ ability for shared representations is not well understood was repeated several times. For instance, the paragraph from 142-157 could be rewritten to be much clearer and more concise. Now the same information is repeated for collaborative experiments and stag-hunt, when the relevant information from these studies is very similar. Also, this is likely a copy-paste error, but the paragraph in lines 536-549 is repeated in lines 550-563.

See above: Reviewer 1, point 2. We have also removed the repeated lines from the copy error

Object vs partner

After reading the manuscript and watching the supplementary material, I wonder to what extent the chimpanzees truly experienced the object condition as being devoid of human intervention. I understand why the current set-up was chosen, and I think it is a clever set up. However, I am not entirely sure if the mere presence of the human observer does not lead to the chimpanzee assuming the human had some kind of role in the flipping of the see-saw, even if they don’t see the exact mechanism. After all, in a zoo setting, chimpanzees are well-acquainted with humans making things happen of which they do not fully understand the mechanism and the causality is not clear (e.g., a caretaker pressing a button to open a hatch far away), so in that sense, I would not be surprised if in both conditions the chimpanzee assumes the human did something to flip the see-saw.

See above: Reviewer 1, point 7

The difference in performance between the object and partner condition, with chimpanzees being more likely to copy the experimenter’s action in the partner condition, could possibly also be attributed to local enhancement. The chimpanzees observe the human experimenter touching the side of the see-saw, and as a result, are more likely to interact with this specific location themselves, resulting in the flip. This alternative would be less cognitively complex than chimpanzees representing their human partner’s mental state, and I think it warrants some mention in the discussion section.

See above: Reviewer 1, point 8

Specific comments line by line

15: So here I would add “cross-species” in front of co-representation, to be very clear.

See above: Reviewer 1, point 1

79: Is the partner in the Joint Simon studies presented here a conspecific or a heterospecific?

The partner in these studies was a conspecific. This has now been clarified:

Line 81 “These studies employed a ‘Joint Simon’ paradigm, wherein subjects must respond to one cue, with a conspecific partner present who must respond to a different cue (18,21).”

225: In what context was the female experimenter familiar? This warrants some more information. Did the subjects have experience with the experimenter providing food? Or was she a neutral observer to them? If food was given in the past, had all chimpanzees experienced this?

See above: Reviewer 1, point 3

271-274: This section was vague to me, and I do think it is important information being shared here that now does not come across. If I understood correctly, trials in the blocked condition did continue if a conspecific of the same experimental condition was present, right? I would re-word this to be a) entirely clear in which situations trials continued with others present b) include justification why you made the decision to continue or interrupt some trials with an audience but others not, and c) in the results, I’d expect to see some information in how many trials conspecifics were present in the trial types where you did allow this. I think you probably have good reasons for deciding on this protocol, but as a reader it is not clear on what grounds these decisions were made, when I think this is relevant information.

See above: Reviewer 1, point 4

446: how many subjects never succeeded?

See above: Reviewer 1, point 5

516: There’s a punctuation mark

---

## [Decision Letter · Decision Letter 1]

14 May 2025

Do chimpanzees (Pan troglodytes) mentally represent collaboration?: Action-learning and communication in a partnered task

PONE-D-25-05158R1

Dear Dr. McEwen,

We’re pleased to inform you that your manuscript has been judged scientifically suitable for publication and will be formally accepted for publication once it meets all outstanding technical requirements.

Kind regards,

Miquel Llorente, PhD

Academic Editor

PLOS ONE

Reviewers' comments:

Reviewer's Responses to Questions

**Comments to the Author**

Reviewer #1: All comments have been addressed

Reviewer #2: All comments have been addressed

2. Is the manuscript technically sound, and do the data support the conclusions?

Reviewer #1: Yes

Reviewer #2: Yes

3. Has the statistical analysis been performed appropriately and rigorously?

Reviewer #1: Yes

Reviewer #2: Yes

4. Have the authors made all data underlying the findings in their manuscript fully available?

Reviewer #1: Yes

Reviewer #2: (No Response)

5. Is the manuscript presented in an intelligible fashion and written in standard English?

Reviewer #1: Yes

Reviewer #2: Yes

Reviewer #1: (No Response)

Reviewer #2: (No Response)

**Do you want your identity to be public for this peer review?** For information about this choice, including consent withdrawal, please see our Privacy Policy

Reviewer #1: **Yes: ** Zoë Goldsborough

Reviewer #2: No

---

## [Editor Report · Acceptance letter]

PONE-D-25-05158R1

PLOS ONE

Dear Dr. McEwen,

I'm pleased to inform you that your manuscript has been deemed suitable for publication in PLOS ONE. Congratulations! Your manuscript is now being handed over to our production team.

Kind regards,

on behalf of

Dr. Miquel Llorente

Academic Editor

PLOS ONE